# Non-Invasive Pulsatile Shear Stress Modifies Endothelial Activation; A Narrative Review

**DOI:** 10.3390/biomedicines10123050

**Published:** 2022-11-28

**Authors:** Jose A. Adams, Arkady Uryash, Jose R. Lopez

**Affiliations:** 1Division of Neonatology, Mount Sinai Medical Center, Miami Beach, FL 33140, USA; 2Department of Research, Mount Sinai Medical Center, Miami Beach, FL 33140, USA

**Keywords:** pulsatile shear stress, whole body periodic acceleration, exercise, enhanced external counterpulsation, whole body vibration, endothelial activation, nitric oxide, cytokines

## Abstract

The monolayer of cells that line both the heart and the entire vasculature is the endothelial cell (EC). These cells respond to external and internal signals, producing a wide array of primary or secondary messengers involved in coagulation, vascular tone, inflammation, and cell-to-cell signaling. Endothelial cell activation is the process by which EC changes from a quiescent cell phenotype, which maintains cellular integrity, antithrombotic, and anti-inflammatory properties, to a phenotype that is prothrombotic, pro-inflammatory, and permeable, in addition to repair and leukocyte trafficking at the site of injury or infection. Pathological activation of EC leads to increased vascular permeability, thrombosis, and an uncontrolled inflammatory response that leads to endothelial dysfunction. This pathological activation can be observed during ischemia reperfusion injury (IRI) and sepsis. Shear stress (SS) and pulsatile shear stress (PSS) are produced by mechanical frictional forces of blood flow and contraction of the heart, respectively, and are well-known mechanical signals that affect EC function, morphology, and gene expression. PSS promotes EC homeostasis and cardiovascular health. The archetype of inducing PSS is exercise (i.e., jogging, which introduces pulsations to the body as a function of the foot striking the pavement), or mechanical devices which induce external pulsations to the body (Enhanced External Pulsation (EECP), Whole-body vibration (WBV), and Whole-body periodic acceleration (WBPA aka pGz)). The purpose of this narrative review is to focus on the aforementioned noninvasive methods to increase PSS, review how each of these modify specific diseases that have been shown to induce endothelial activation and microcirculatory dysfunction (Ischemia reperfusion injury-myocardial infarction and cardiac arrest and resuscitation), sepsis, and lipopolysaccharide-induced sepsis syndrome (LPS)), and review current evidence and insight into how each may modify endothelial activation and how these may be beneficial in the acute and chronic setting of endothelial activation and microvascular dysfunction.

## 1. Introduction

The luminal surface of all the vasculature and the heart is lined by endothelial cells (EC), encompassing more than 5000 m^2^. Furthermore, the response of EC to external signals and the synthesis and production of various mediators is heterogeneous and adaptive based on location and signals [1,2,3,4,5,6,7,8,9,10,11]. EC membranes are the sensing mechanism, responsive to mechanical (shear stress) and biochemical signaling (chemosensor) [6,12]. EC output is important for blood fluidity, coagulation, vasoreactivity, vasculogenesis, barrier function, and inflammation [13]. Endothelial cell activation is the process by which EC changes from a quiescent cell phenotype, which maintains cellular integrity, antithrombotic, and anti-inflammatory properties, to a prothrombotic, pro-inflammatory, and permeable phenotype, also at the site of injury or infection, involved in repair and leukocyte trafficking. Endothelial activation is triggered by a multitude of stimuli that include inflammatory cytokines (interleukins, tumor necrosis factor, and interferon-γ), bacterial endotoxins, and pattern recognition receptor activation (PRR) after recognition of pathogen-associated molecular patterns (PAMP) or damage-associated molecular patterns (DAMP) [14,15,16]. Pathological activation of EC leads to increased vascular permeability, thrombosis, and an uncontrolled inflammatory response leading to endothelial dysfunction; the latter can be contained at the local level or participate in a more profound systemic response leading to multiorgan dysfunction and death [14,17,18,19,20,21,22].

The sensing capabilities of EC include tangential, radial, axial, pulsatile, and oscillatory shear flow patterns [23,24]. Pulsatile shear stress (PSS) is a periodical laminar flow that occurs predominantly in the straight part of the blood vessel; it has a positive mean flow rate and fluid velocity that oscillates at the frequency of the heart rate [25,26,27,28]. PSS promotes a cytoprotective, non-inflammatory EC phenotype and vascular health [20,24], in contrast to oscillatory shear (mainly seen in arterial bifurcations and curvatures and where blood flow is uneven), which promotes an inflammatory and atherogenic phenotype [28,29,30,31,32,33,34,35].

The balance between vasoconstriction/relaxation is also governed by EC through vasoconstriction mediators (endothelin-1 and thromboxane-A2) and vasodilators (nitric oxide (NO), prostacyclin, and endothelium-derived hyperpolarizing factor (EDHF)). Nitric oxide (NO) is a gas that is important not only for vasodilation, but also critical in signal transduction [36,37,38,39]. NO production occurs through the oxidation of L-arginine to L-citrulline using the enzyme nitric oxide synthase (NOS) and tetrahydrobiopterin (BH4). There are three nitric oxide synthases: (a) endothelial nitric oxide synthase derived from the endothelium (eNOS also called NOS3) is constitutively expressed, calcium-dependent, and produces nanomolar amounts of NO and produced in response to PSS; (b) inducible nitric oxide synthase (iNOS also called NOS2) is not constitutively expressed, produces large quantities of NO, and is usually produced by macrophages and inflammatory cells; (c) neuronal nitric oxide synthase (nNOS also called NOS1) is found mostly in both neuronal and cardiovascular tissue and has a role in neuronal signal transduction and chronotropicity of the heart [37,40].

Experiments performed in the early 1990s by Hutchenson et al. found that NO is produced by EC as a function of pulsatility with an optimal frequency of pulsation of 2–8 Hz (120–480 cpm) [41]. PSS activates eNOS through phosphorylation, thus producing eNO, which is important for increasing blood flow and an important signaling molecule that down-regulates the inflammatory cascade and improves microvascular dysfunction [37,42,43,44,45,46,47,48,49]. eNO also serves to counterbalance signals that mediate endothelial activation [50].

The baseline pulsations in the human circulation are in the frequency range of 1 to 2 Hz and additional pulsations beyond these increase NO bioavailability via eNOS [41,51]. The concept of modulating the endothelium using noninvasive methods that induce PSS is a novel paradigm shift in thinking. In this narrative, we will focus on the modification of endothelial activation using noninvasive PSS, of which the archetype is exercise. Therefore, we review simple methods for inducing PSS under conditions known to produce endothelial cell activation.

This review will focus on four noninvasive and nonpharmacologic methods to increase PSS that have been shown to confer a protective role in ischemia reperfusion injury (IRI), sepsis or Escherichia Coli endotoxemia, lipopolysaccharide-induced sepsis-like syndrome (LPS), exercise (EXER), enhanced external counterpulsation (EECP), whole-body vibration (WBV), and whole-body periodic acceleration (WBPA, also known as, pGz) (Figure 1).

## 2. Methods

A review of the literature on endothelial activation was carried out using the following databases: EMBASE, PUBMED, SCOPUS. Each database was searched for the following key words [Endothelial activation] or [endothelium] and one of the three modalities of inducing noninvasive PSS: Exercise, WBV, WBPA, or pGz. The latter combination was combined with either; (a) sepsis, infection or inflammation, or lipopolysaccharide (LPS), or (b) ischemia reperfusion or myocardial infarction or cardiac arrest or resuscitation or post-resuscitation injury (Appendix A contains search strategy and the number of articles). The studies were limited to those in English. The titles and abstracts were reviewed for relevant information on the various interventions of pulsatile shear stress in relation to disease models. Search dates: January 1990 to September 2022.

## 3. Models of Endothelial Activation

### 3.1. Ischemia Reperfusion Injury (IRI)-Cardiac Arrest (CA) and Myocardial Infarction (MI)

Estimates indicate that in the United States there are more than 500,000 cases annually of outpatient and in-hospital cardiac arrest, with a return to spontaneous circulation (ROSC) of 40–50% and a survival rate to hospital discharge of 10.5% and 26.7% for outpatient and in-hospital cardiac arrest, respectively [52]. Postcardiac arrest syndrome (PCAS) is characterized by reperfusion injury from systemic ischemia, myocardial dysfunction, brain, and other vital organ injury, superimposed on underlying diseases, all of which explain the low survival rate of hospital discharge. After CA, a systemic inflammatory response occurs that has been shown to occur in the reperfusion stage, with the release of pro-inflammatory cytokines by leukocytes and endothelial cells through the activation of leukocytes and endothelial cells and the release of secondary cytokines [53,54,55,56,57,58,59,60]. EC expresses a wide spectrum of cytokines and chemokines, including pro-inflammatory interleukins; IL-1β, IL-3, IL-5, IL-6, IL-8, IL-11, IL-15, and tumor necrosis factor (TNF-α), as well as anti-inflammatory cytokines such as IL-1 receptor antagonist (IL-1ra), IL-10, IL-13, and transforming growth factor beta (TGF-β) [61,62,63].

Recent data in mice show that myocardial infarction (MI) produces remote global endothelium activation, with up-regulation of the vascular cell adhesion molecule (VCAM-1), the cell adhesion molecule (P-Selectin) and platelet adhesion in remote arterial and microvascular beds, which persists for longer periods in animals with preexisting atherosclerosis [64,65]. Several reviews have outlined the role of NO in the cardiovascular system [47,48]. CA and resuscitation are models of total-body IRI, while MI with and without reperfusion are models of focal injury. These two models offer the opportunity to study what has been termed “sterile” endothelial activation [66].

### 3.2. Sepsis and Lipopolysaccharide-Induced Sepsis Syndrome (LPS)

In 2017, the estimated global burden of sepsis was 48.9 million people worldwide, with a mortality of 11 million. In the United States, for example, sepsis is the most common cause of in-hospital deaths and costs more than USD 2 billion annually [67]. Sepsis is a dysregulated response to infection that triggers a complex set of pathways and a cellular response that includes endothelial activation, macro, and micro circulatory failure, ultimately leading to organ failure and death. The systemic response may be triggered by the recognition of a pathogen (Pathogen Associated molecular pattern, PAMP) and or cellular injury proteins (damage associated molecular pattern, DAMP) recognition. The initial response predominantly by immune cells is the release of cytokines, interleukins, chemokines, interferons, tumor necrosis factor (TNF-α), and growth factors. The initiation of the inflammatory cascade is orchestrated by nuclear factor kappa beta (NFƙ-β). The target of cytokines is the EC; however, these are also capable of secreting cytokines. During sepsis, the role of EC is to amplify the immune response and activate the coagulation system, with endothelial activation ultimately contributing to end organ damage and microcirculatory failure [66,68,69]. In endotoxin (Escherichia Coli) -induced lipopolysaccharide sepsis-like syndrome (LPS), bacterial cell wall products ultimately bind to the toll-like receptor-4 (TLR-4) on the endothelial cell wall, to induce the intracellular response of EC of cytokines, adhesion molecules, and reactive oxygen species (ROS), and similarly amplify the immune response. Elevated biomarkers of endothelial activation/dysfunction in the systemic inflammatory response of critical illnesses and sepsis have been shown to be associated with a higher risk of developing respiratory failure, multiple-organ dysfunction, and death [18,70,71,72]. The decrease in NO bioavailability also plays a role in sepsis. Systemic NO release during sepsis has been shown and later thought to be responsible for hemodynamic and vascular instability, prompting the use of a non-selective inhibitor of NO (L-NAME) as a clinical therapeutic intervention in sepsis, which failed [73,74]. In animal models, an increase in NO bioavailability using NO donors or arginine administration, or a decrease in asymmetric dimethylarginine (ADMA, an endogenous inhibitor of nitric oxide synthase) appears to show some promise [75]. It is important to note that the reduced bioavailability of NO at the microvascular level comes primarily from the reduction in eNOS produced by the EC and that produces small amounts of NO, in contrast to iNOS which produces large amounts of NO, primarily by neutrophils and macrophages. NO derived from eNOS has been shown to be protective in sepsis [76,77]. Thus, increasing the bioavailability of NO through eNOS would provide a new avenue of therapeutics [46,48,78].

## 4. Exercise

### 4.1. Exercise for Pulsatile Shear Stress

Exercise is defined as “a subset of physical activity that is planned, structured, and repetitive and has as a final or intermediate objective the improvement or maintenance of physical fitness” [79,80]. For this narrative review, the exercise strategies considered involve walking, jogging, or running. PSS and/or circumferential wall stress or stretch (which arises from the effect of blood pressure on the vascular wall and is applied to all layers of the arterial wall) are the primary signals produced by EXER that are mechanically transduced by the EC [8,29,34,81,82,83,84,85,86].

During walking, jogging, or running, pulses are added to the circulation as a result of the frequency of cadence of the foot. The frequency of steps for both men and women who are recreational runners has been estimated to be between 163 and 169 steps per min [87]. This frequency, added to a baseline pulsatility of 60–100 beats/min, generates total pulsation close to 220–240 beats per min (3–4 Hz) during running. Since running is not timed for the cardiac cycle, the expected pulsatility frequencies are in the range of 1.3 to 4 Hz.

This narrative review will not discuss the various exercise strategies or mechanisms of exercise-induced cardioprotection that others have thoroughly reviewed [88,89,90,91,92,93,94,95,96,97,98]. Exercise reduces cardiovascular morbidity and mortality and is positively correlated with beneficial health outcomes but requires subject cooperation and thus may prevent patients from participating and remaining in an exercise program, particularly those in an intensive care setting or those with physical and cognitive limitations.

### 4.2. Exercise (EXER) and Ischemia Reperfusion Injury

Exercise is a well-known cardioprotective strategy. Physical activity, exercise, and a healthy diet are the pillars of cardiovascular health [99]. Exercise induces a variety of cardioprotective signals, including a decrease in the inflammatory phenotype [98]. Regular exercise induces interleukin-6 (IL-6) produced by muscle fibers, which stimulates anti-inflammatory cytokines (IL-1ra and IL-10) and inhibits tumor necrosis alpha (TNF-α) [100]. In a recent systematic review and meta-analysis, regular exercise decreased aging-induced inflammasome activation related to inflammatory cytokines (IL-1β and IL-18) [101]. In exercised (voluntary free-wheel running) mice fed a high-fat diet, exercise suppressed the pyrin domain of the NOD-like receptor family containing the 3 (NLRP3) inflammasome, improved nitric oxide production, and reduced oxidative stress [102].

Two specific periods have been explored concerning the role of EXER in cardiovascular protection against IRI or MI: EXER as a preconditioning (pretreatment) strategy (EXER performed prior to the onset of IRI) and a postcondition (post-treatment) strategy (EXER, performed after IRI). Both strategies aim to induce cardioprotection through various pathways, which ultimately increase myocardial tolerance, reduce the size of the infarct, and IRI-induced arrythmias. The effects of exercise preconditioning against IRI have been well-established in animal models and human epidemiological studies summarized by Borges et al. [103]. Additionally, the beneficial effects of exercise after MI have also been well established, with exercise being an important component of the post-MI rehabilitation program [104]. However, a link between cardioprotection induced by exercise and decreased endothelial activation remains to be established. Exercise has been shown to increase the bioavailability of NO, specifically eNO and IL-6, both of which play an anti-inflammatory role [105,106,107]. It is important to note that excessive, prolonged, and strenuous overtraining can lead to damaging oxidative stress, with an attendant decrease in NO bioavailability. The concept of redox and exercise-induced hormesis has been advanced and previously reviewed; therefore, it appears that too much of a good thing may not necessarily be effective when it comes to EXER [80,108,109,110,111,112].

Data on the beneficial effects of EXER as a pre- or post-conditioning strategy for cardiac arrest and resuscitation are scarce. There is little doubt about the cardioprotective role of EXER and physical activity, on overall cardiovascular health suggesting a beneficial effect of EXER, and the benefits of post-MI EXER rehabilitation [113,114,115]. Recent data from the Korean National Outpatient CA registry showed that patients with higher intensity physical activity during exercise before and index CA had better survival outcomes and a successful percutaneous coronary intervention [116] suggesting a protective role for EXER specifically in CA.

### 4.3. Exercise and Sepsis

The effects of exercise on survival in animal models of sepsis and LPS have been well-documented, showing a favorable survival response to various EXER interventions [117,118,119,120,121,122,123,124].

Gholamnezhad et al. recently performed a systematic review of the modulatory effects of EXER on LPS-induced lung inflammation. The results showed that aerobic exercise (prior to LPS) in rodents reduced LPS-induced oxidative stress, inflammation, protein leakage, levels of IL-6, IL-1β, IL-17, TNF-α, granulocyte–macrophage colony stimulating factor, and improved IL-10 and IL-1Ra, and a change in the balance between pro-inflammatory and anti-inflammatory phenotype, thus supporting the role of exercise in LPS-induced lung injury [125].

In human studies, the effects of exercise on sepsis have also been reported in a limited number of studies. Low rates of physical exercise and high rates of watching television (physical inactivity) are associated with higher morbidity and mortality from community-acquired sepsis [109], and physical rehabilitation in septic patients was shown to improve physical function and reduce the inflammatory response [126,127]. A review on the effects of exercise in the treatment of sepsis in animal models and patients has recently been published [128].

## 5. Enhanced External Counterpulsation

### 5.1. Enhance External Counterpulsation (EECP) for Pulsatile Shear Stress

Enhanced external counterpulsation (EECP) involves compression of the legs to buttocks using pneumatic cuffs, timed to early diastole [129,130]. EECP induces pulsations and imparts a circumferential stretch that doubles the heart rate (2–3.3 Hz). The beneficial clinical effects of EECP have been reported for angina, peripheral artery disease, diabetes, erectile dysfunction, and possibly Alzheimer’s disease. Similarly, to exercise, EECP via PSS induces NO production [131], improves endothelial function [132,133], and attenuates pro-inflammatory signaling pathways [134,135]. The risks and guidelines of EECP have been published by Lin et al. [136].

### 5.2. Enhanced External Counterpulsation (EECP) and Ischemia Reperfusion Injury

The use of EECP on acute and chronic MI-induced IRI has been shown in both animal models and human studies and has been reviewed [129], and its use as a myocardial conditioning strategy has also been reviewed [137].

In nonischemic hypercholesterolemic patients, seven weeks of EECP was compared with the control. The EECP group had a significant reduction in atherosclerosis lesion, and a reduction in C-reactive protein, vascular cell adhesion molecule-1(VCAM-1), iNOS, mitogen-activated protein kinase phosphorylation (MAPK-p38), and activation of NFƙ-β [134].

In a dog model of MI by coronary artery occlusion, EECP use significantly improved myocardial perfusion and function after 4 and 6 weeks of EECP compared to controls, along with increased expression of vascular endothelial growth factor (VEGF) and increased microvascular density [138]. EECP has also been studied in a dog model of CA. Post-CA EECP (4 h of use after CA) increased cerebral blood flow, improved microcirculation recovery, and improved neurological outcomes from 24 to 96 h compared to control animals [139]. In a similar dog model of CA with also 4 h of EECP post-CA, others have also shown improved survival and myocardial function [140].

Casey et al. and Braith et al. studied the effects of EECP in patients with chronic angina and symptomatic coronary artery disease, respectively. Both investigators showed a significant reduction in TNF-α, monocyte chemoattractant protein-1 (MCP-1), and soluble vascular adhesion molecule (sVCAM-1) compared to controls [133,141]. Yang et al. have summarized the additional benefits of EECP beyond hemodynamics [142].

### 5.3. Enhanced External Counter Pulsation and Sepsis

The effects of EECP on sepsis or LPS animal models have not been published. A single case study has documented the use of EECP in a female patient diagnosed with coronavirus disease 2019 (COVID-19) and treated at home with a cocktail of vitamins and hydroxychloroquine for 6 days. Most of her symptoms resolved but remained with fatigue, headaches, and shortness of breath at rest and during activities, and “brain fog” (subjective lack of clarity) for months. She was treated with 35 sessions of EECP, and the aforementioned symptoms resolved. The resolution of symptoms was subjectively measured, and this resolution was attributed to the use of EECP [143].

## 6. Whole-Body Vibration

### 6.1. Whole-Body Vibration (WBV) for Pulsatile Shear Stress

The effects of vibration on the entire body were first described in the 1900s and began to appear in the scientific literature in the 1960s describing the effects of whole-body vibration (WBV) on ventilation, behaviors, and central hemodynamics [144,145,146,147].

The mechanical oscillations imparted by WBV are performed using a platform, moving in a linear or pivotal motion with a standing or seated subject at frequencies from 12–60 Hz and displacements from 1 to 10 mm producing accelerations of + 1.5 mt/sec^2^ [148,149,150,151,152]. Most studies exploring the effects of WBV use a structured exercise program performed on the WBV platform. WBV has been shown to increase skin blood flow [153,154,155], improve endothelial function in an elderly population [156], and its effect is summarized by others [157,158,159].

### 6.2. Whole-Body Vibration and Ischemia Reperfusion Injury

WBV has been used as a pretreatment strategy in a rodent model of acute MI without reperfusion. WBV was carried out for 1 and 3 weeks (30 min per day for 6 days) versus a control group. Myocardial infarct size and severity of ventricular fibrillation were significantly lower in WBV at 1 and 3 weeks [160]. WBV has also been used as an adjunct to a cardiac rehabilitation program over a 24-days, both standard exercise rehabilitation and those who received adjunct WVB had improvement in exercise tolerance and left ventricular ejection fraction, and both groups obtained similar effects [161]. There are no published results on the use of WBV post CA, likely due to the need for subject cooperation and the critical nature of these patients.

### 6.3. Whole-Body Vibration and Sepsis

Similarly, to EXER, data on WBV applied in the setting of sepsis or LPS are scarce. A single study explored the effects of WBV on LPS-induced inflammatory bone loss. The report focused on trabecular bone loss, which decreased with WBV. Furthermore, in the same study using in vitro stimulation of human mesenchymal stromal cells with WBV, WBV reduced the increase in LPS-induced IL-1β and TNF-α induced by LPS [162]. Two recent reviews have addressed the potential for WBV to improve acute and long-term clinical conditions associated with COVID-19 and provide a framework for the use of WBV in the acute care setting [163,164].

Sanni et al. investigated the acute effects of WBV (HI = 88.7 ms^2^ and LO =44.4 ms^2^, both at 30 Hz) in healthy volunteers. They report a higher muscle oxygen consumption in the LO compared to the HI group and an increase in IL-6 in both groups but a higher in the LO [165]. Rodriguez-Miguelez studied healthy elderly volunteers (70 years) who performed an 8-week training protocol on WBV. WBV produced a significant decrease in TNF-α, and an increase in IL-10, and a significant decrease in the mediator myeloid differentiation response gene88 (MyDD88, an essential protein in the production of inflammatory cytokines) and transcription factor p65 (also known as the nuclear factor NF-β p65 subunit) [166]. In contrast, Jawed et al. showed in healthy male volunteers trained on WVB (35 Hz, eight 60 s sets, with 2 min between sets) an increase in IL-10 and an increase in TNF-α [167]. Similarly, Neves et al., in adult patients with chronic obstructive pulmonary disease (45–80 years) enrolled in a 12-week WBV protocol, and Cristi et al., in a 9-week WBV protocol in elderly volunteers (80 years), failed to show changes in IL-6 or soluble receptors of TNF, TNF-α, IL-10, and IL-1β [168,169]. Therefore, the effects of WBV on the parameters of inflammation and endothelial activation markers remain inconsistent.

## 7. Whole-Body Periodic Acceleration (WBPA aka pGz)

### 7.1. Whole-Body Periodic Acceleration (WBPA, aka pGz) and Pulsatile Shear Stress

WBPA is the sinusoidal motion of the body on a platform in a headward to footward direction, which via inertial forces introduces small pulsations to the vasculature inducing PSS. [170,171]. The pulsations produced by WBPA are not synchronized with the cardiac cycle. The WBPA motion frequency in humans is between 100 and 150 cycles per min (1.6 to 2.5 Hz) with acceleration forces in the z plane Gz ± 0.3 mt/sec^2^. A more detailed description of the platform has previously been reported [147,170,171,172]

WBPA has been shown to produce endothelial-derived NO release in human subjects [170,173,174], and animal models [171,175,176] and genomic upregulation of eNOS occurring over a relatively short time period [175,177], along with increased expression of antioxidant capacity [177], and endothelial-derived anticoagulant, vasoactive proteins, and adrenomedullin [172,178].

### 7.2. Whole-Body Periodic Acceleration (WBPA, aka pGz) and Ischemia Reperfusion Injury

The use of WBPA in CA models was first evaluated nearly 20 years ago, when laboratory observations on its use as a cardiopulmonary resuscitation (CPR) method were made [179]. In an adult porcine model of CA (15 min of ventricular fibrillation), WBPA used as a CPR method allowed for successful resuscitation and return of circulation in 100% of animals with normal neurological evaluation 24 h later [179]. This observation was also replicated in a newborn porcine model of CA induced by asphyxia [180]. The use of WBPA as a CPR method can be considered a method of perconditioning (or conditioning), which is an intervention performed during the ischemic period designed to improve injury produced by ischemia and reperfusion [137,181].

WBPA used before CA (preconditioning) has also been used in CA porcine models. In addition to a decrease in post-resuscitation myocardial stunning and improved regional blood flow to vital organs, WBPA increases eNOS expression in the myocardium and decreases tissue damage [182]. WBPA used as a preconditioning strategy in a rat model of MI improved survival, decreased infarct size, and increased eNOS expression [183].

WBPA has also been used after CA (postconditioning) in the same porcine model of ventricular fibrillation of CA, showing decreased post-arrest myocardial stunning, improved regional microvascular blood flow to vital organs, and increased expression of eNOS [184]. Chronic use of WBPA (1 h daily for 4 weeks) has also been used in a rat MI survival model as a chronic postconditioning strategy. WBPA improved survival and myocardial contractility and decreased myocardial fibrosis compared to controls. Furthermore, WBPA restored the expression of eNO, decreased the expression of IL-6 and TNF-α, and increased the expression of IL-10 [185]. The role of endothelium and nitric oxide synthases in CA and resuscitation was previously reviewed, showing the importance of eNO [169]. Together, these studies suggest that PSS induced by WBPA is cardioprotective, reduces the inflammatory phenotype induced by CA and focal MI, and improves microvascular flow to vital organs.

Currently, there are no human studies on the use of WBPA in patients with CA. In patients with angina, who are not candidates for percutaneous coronary intervention and/or coronary artery bypass surgery, WBPA used for 20 days improved exercise capacity and improved myocardial perfusion and function [186]. In patients with severe leg ischemia, WBPA performed for 10 days significantly increased laser Doppler blood flow to the ischemic limb, and these findings were also replicated in a mouse model of severe leg ischemia, which also confirmed upregulation of eNOS and VEGF expression [187].

### 7.3. The Effects of WBPA/pGz on Sepsis

WBPA was used as a pre- and post-treatment strategy in a murine lethal dose endotoxemia (LPS) model. Both WBPA strategies have been shown to increase survival, decrease microvascular leakage, and restore the expression of the tunica interna endothelial cell kinase-2 (TIE2) receptor enriched with tyrosine kinase and its phosphorylation (important for maintaining tight junctions of the endothelium and decreasing vascular permeability) [188]. Furthermore, WBPA restored the expression of eNOS and decreased the proinflammatory cytokines; TNF-α, NFƙ-βp65, IL-1β, and IL-6, and increased the anti-inflammatory cytokine Il-10 [189]. These studies suggest that WBPA through PSS has a protective effect (preconditioning and postconditioning) effect on LPS-induced endothelial activation and improves the pro-inflammatory environment induced by LPS. The effect of WBPA on human subjects with critical illness or sepsis has not been explored. It should be noted that PSS (due to the antiviral properties of NO) has been theorized to potentially be therapeutic adjuncts in the fight against COVID-19 [190].

The WBPA human platform is large and heavy; therefore, a smaller and simpler device (a predicate device for WBPA) was designed. The details of this passive simulated jogging device (JD, (Gentle Jogger, Movewell Technologies LLC, Hollywood, Florida, USA)) have been previously published [191,192,193,194]. The JD passively moves the feet using a motor, in alternating motion simulating walking or jogging, and with each downward stroke of the fore foot, a pulsation is added to the circulation, thus inducing PSS. This device has been shown to increase NO bioavailability in humans [195].

The characteristics of the four described interventions to produce PSS have been previously summarized [147] (Figure 2). These vary with respect to frequency, accelerations, and known effects on endothelial activation. All have been shown to increase the bioavailability of eNOS and NO, improve endothelial function, and induce an anti-inflammatory and fibrinolytic effect. They are also varied in their subject–device interphase and portability, and thus the limitation of usage particularly in critically ill patients, and those with physical or cognitive limitations.

## 8. Limitations

The review has focused on four of the most published methods for PSS; however, there are other methods. Vibroacoustic Therapy (VAT) is a non-invasive delivery of a sound frequency of 50–100 Hz and a sound pressure of 0.5–20 dyn/cm^2^ by a sound speaker that delivers stimulation directed toward the vascular wall. Sound-distributed stimulation to the endothelial cell matrix can trigger mechanosensors in cells affected by PSS and has been shown to increase eNO, can induce the expression of the mechanosensory protein Syndecan-4 (Syn4) and VEGF, as well as increase blood flow and clot dissolution [196,197,198,199,200]. This method may also be a plausible therapeutic strategy for MI [201,202] and sepsis [203].

Additional methods of delivering PSS also include passive cycling or limb movement [204,205], as well as a recently published method of 160-degree V-shaped WBPA in supine posture [206]. The latter methods may also have potential for clinical therapeutics, but studies on their use in MI, CA, or sepsis are not available. The study of endothelial activation markers in the context of these PSS interventions is an open area of investigation that may provide fruitful areas of mechanistic research and therapeutic interventions in a variety of diseases.

## 9. Conclusions

The effects of PSS on the vascular endothelium provide a protective and potential therapeutic intervention for the management of diseases with acute or chronic endothelial activation, endothelial dysfunction, and reduced microvascular flow. This narrative review has shown that externally applied PSS, as provided by EXER, EECP, WBV, and WBPA/pGz, positively modifies endothelial activation, improves microvascular dysfunction, and can be a viable adjunct as a clinical intervention. More studies are needed to improve the subject device interface, provide guidance on the optimal duration of use of these devices, and elucidate additional mechanistic insights.

## Figures and Tables

**Figure 1 biomedicines-10-03050-f001:**
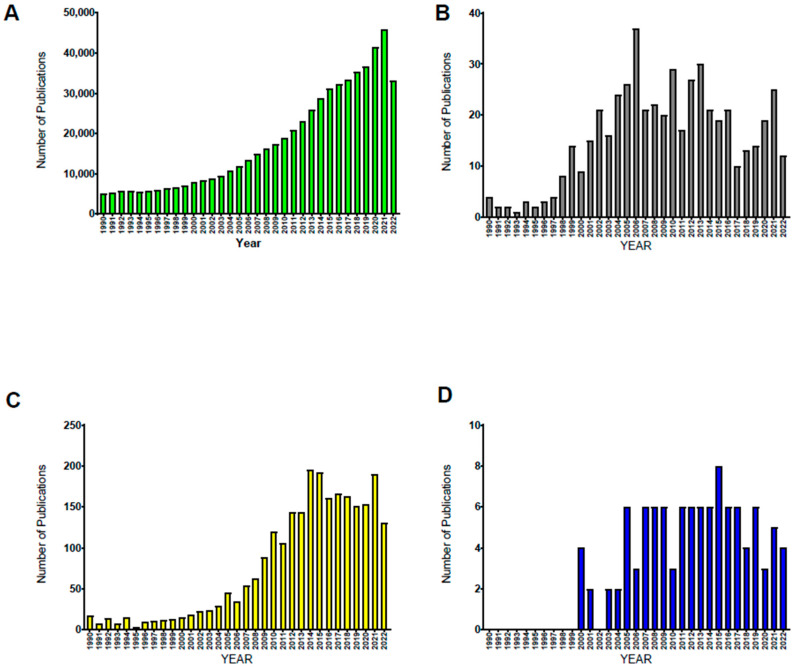
***Publications in PUBMED from 1990 to 2022 for Four Methods of Noninvasive Pulsatile Shear Stress.*** Number of publications in PUBMED by year for the four methods described in this manuscript of noninvasive pulsatile shear stress: (**A**) Exercise, (**B**) Enhanced External Counterpulsation, (**C**) Whole-Body Vibration, (**D**) Whole-Body Periodic Acceleration (also known as pGz).

**Figure 2 biomedicines-10-03050-f002:**
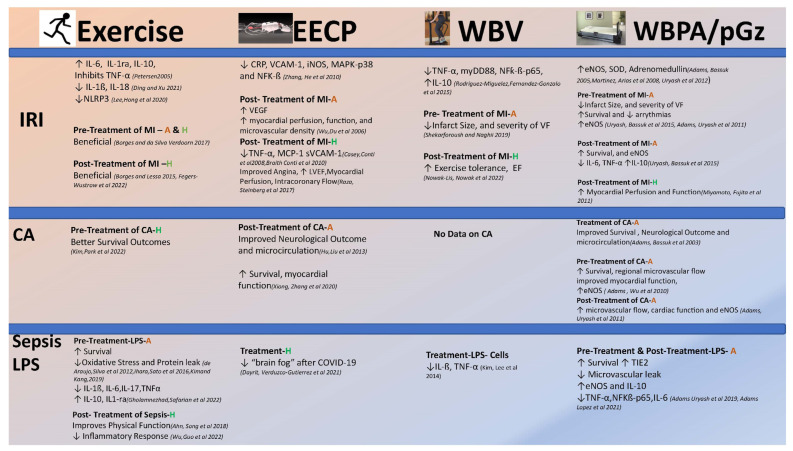
Summary of the Effects of Exercise, Enhanced External Counterpulsation (EECP), Whole-Body Vibration (WBV), and Whole-Body Periodic Acceleration (WBPA also known as pGz) on Ischemia Reperfusion Injury (IRI), Cardiac Arrest (CA), and E. Coli endotoxin induced Sepsis-like syndrome (LPS). Summary table of the effects of four noninvasive methods to induce pulsatile shear stress (PSS), and conditions, that increase endothelial activation and decrease microvascular flow; Ischemia Reperfusion Injury (IRI), cardiac arrest (CA), and sepsis/E. Coli endotoxin induced sepsis like syndrome (LPS). For each intervention, a summary of the changes is provided, when the method to induce PSS is performed as a pretreatment (preconditioning) or posttreatment (postconditioning) strategy, in A (Animals), H (Humans). Abbreviations: ↑ = increase, ↓ = decrease, MI = Myocardial Infarction, VEGF = vascular endothelial growth factor, TNF-α = tumor necrosis factor alpha, IL-1β = interleukin1-beta, IL-6 = interleukin 6, IL-17 = interleukin 17, IL-10 = interleukin 10, IL1-ra = interleukin 1 receptor antagonist, MCP-1 = Monocyte Chemoattractant Protein-1, sVCAM-1 = soluble Vascular Cell Adhesion Molecule-1, TIE2 = endothelial-specific receptor tyrosine kinase family, eNOS = endothelial derived nitric oxide synthase, NFƙβ-p65 = nuclear factor kappa beta p65 subunit, VF = ventricular fibrillation, LVEF = left ventricular ejection fraction, EF = Ejection Fraction. [100,101,102,103,114,115,116,117,121,122,125,126,128,129,133,134,138,139,140,141,143,160,161,162,166,172,175,177,178,179,182,183,184,185,186,188,189].

## Data Availability

Not applicable.

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
