# Peer review of "Non-Invasive Pulsatile Shear Stress Modifies Endothelial Activation; A Narrative Review"

_biomedicines, 2022, doi:10.3390/biomedicines10123050_

Round 1

Reviewer 1 Report

In this review, the authors focus on four noninvasive and nonpharmacologic methods to increase pulsatile shear stress. This approach is interesting. This reviewer has a few concerns as below.

Major comments

The authors explained, “This narrative review will not discuss the various exercise strategies or mechanisms of exercise-induced cardioprotection that others have thoroughly reviewed”. However, the different exercise modalities (aerobic, resistance, stretching, etc) obviously affect the results (positive or negative). So, please define “exercise” in this review.

The authors focus on PSS because PSS as mechanical stress increases NO bioavailability due to secretion from EC. Meanwhile, the NO bioavailability is impaired by the increased oxidative stress (negative impact on eNOS via PRMT-ADMA pathway and BH4 oxidation to BH2 in addition to activation of Arginase), which is possibly caused by higher intensity exercise etc. Even though there may be no reference to IRI, CA, and Sepsis LPS, exercise can be effective in the right hands but dangerous in the wrong hands. So, these phenomena should be considered/mentioned in the text to avoid misreading.

This reviewer could not follow this conclusion from the data set in this review. Why can the authors lead this conclusion? The conclusion should similar to the purpose.

Minor comments

L18; Please add space between “(PSS)” and “are”.

L20; Please remove the spell-out about PSS because of duplication.

L38; Please remove “cell” because “EC cell membranes” means “endothelial cells cell membranes”

L50; Please spell out “PSS”.

L70, L72, L195, and L408; Please correct “eNO” to “eNOS”

Could you please increase the size of both figures? It is hard to read the text in both figures.

Author Response

We thank this reviewer for some insightful comments that have strengthened our narrative review.

Excellent comment on the definition of the exercise, which is now done in the manuscript.

Comments about the increase in oxidative stress leading to a decrease in NO bioavailability by higher intensity exercise, and cautioning the reader on this, are important points. We have added some sentences with references to this issue, lines 212-216. 

The conclusion has been written to capture better the essence of the purpose of this narrative review.

Minor Comments

All comments regarding space, spelling out, and cell removal after EC, have been corrected.

In the text, eNO refers to nitric oxide produced via endothelial nitric oxide synthase (eNOS).

We apologize for the figure sizes; it is a problem with the formatting of the submitted word file. The actual figures are high-resolution and very easy to read.

Reviewer 2 Report

The manuscript should be carefully proof read as there are several misspelling in the text.

Minor comments

1-in the abstract, line 20: PSS has already been defined line 17. So pulsatile shear stress can be removed.

2- in the abstract, the last sentence is not understandable and should be modified.

3-line 49: “;” should be changed for “:”

Major comments

1-the author should define more precisely PSS and oscillatory stress as well as the differences between these 2 shear stresses.

2- paragraph 3.1: the authors should detail more the cytokines that are released by leukocytes and endothelial cells

3-circumferential wall stress should be defined

Author Response

We thank this reviewer for asking us to expand on the differences between shear stress and cytokines expressed by endothelial cells.

We have proofread and spell-checked the manuscript several times.

Major Comments:

As requested, we have more precisely defined PSS and Oscillatory Stress and have referenced these, for further reading. These changes appear on lines 53-59 and 168-174.

Cytokines expressed by EC have also been included in lines 118-122.

Circumferential wall stress is defined and referenced in Lines 171-174.

Minor Comments:

All comments have been addressed.

Reviewer 3 Report

The manuscript entitled:" Non-Invasive Pulsatile Shear Stress Modifies Endothelial Activation; A Narrative Review" focused on the evaluation of pulsatile shear stress, four noninvasive methods to induce pulsatile shear stress and conditions, that increase endothelial activation and decrease microvascular flow. The manuscript is well written and requires minor considerations to be accepted for the publication:

- In the introduction section, the authors report the endothelial cells, endothelial cell activation, but next are absent the information about endothelial cells, how changes it’s from a quiescent cell phenotype, to a prothrombotic, pro-inflammatory, and permeable phenotype. Authors should add information in manuscript.

There is enough information about effects of exercise, enhanced external counterpulsation, whole body vibration and whole body periodic acceleration but little specific data about endothelial cell activation.

Figure 1 should is improved.

Author Response

The authors thank this reviewer for the laudatory comments on the writing of this manuscript.

Comments on endothelial activation are very well taken. We have expanded the Introduction section on this specific subject in lines 41-52 with appropriate references.

We apologize for the figure sizes; it is a problem with the formatting of the submitted word file. The actual figures are high-resolution and very easy to read.

Reviewer 4 Report

This review submitted by Adams JA et al is compact and well organized from a unique perspective and will be of interest to readers. However, the summary figure (Fig. 2) is not easy to understand, so it needs to be made more comprehensible to readers by using illustrations.

Author Response

We thank this reviewer for his/her compliments as to the organization and succinctness of the data presentation, and the unique perspective.

We have tried to make Figure 2 as an illustration (even prior to this submission), however, the Table format presents a lot of information that is referenced. For each of the interventions (EXER, WBV, etc) we show their summarized published effect on IRI, CA, and Sepsis. Furthermore, the intervention has been broken down as to when it took place (pre or post-treatment), and lastly, the data was classified as originating from human (H) or animal (A) studies.  This information is too complex for an illustration. To be very specific we have also referenced the study from which the data were obtained.  The size of the figure is a problem with the formatting of the submitted word file. The actual figures are high-resolution and very easy to read.

Round 2

Reviewer 1 Report

Well done. This reviewer does not have further comments. Congratulations!